# Interpersonal and Intrapersonal Skills for Sustainability in the Educational Robotics Classroom

**David Llanos-Ruiz \*, Vanesa Ausin-Villaverde and Victor Abella-Garcia**

Department of Educational Sciences, Faculty of Education, University of Burgos, C/Villadiego 1,
09001 Burgos, Spain; vausin@ubu.es (V.A.-V.); vabella@ubu.es (V.A.-G.)
\* Correspondence: dlruiz@ubu.es

**Abstract:** Education is an indispensable tool for improving social sustainability. In the school context, a wide variety of methodologies are being considered to achieve this goal by promoting cultural and experiential sustainability through educational and technological innovation. Educational robotics is an educational–formative context that makes it possible to develop new learning environments, enhance sustainable curriculum development, and promote active student participation. The general objective of this research is to analyze the perceptions of teachers of technology, robotics, and/or programming and to study the social benefits of interpersonal, intrapersonal, and/or academic skills of students to improve curricular sustainability during the teaching–learning process from the perspective of robotics and programming in students in early childhood education, primary education, compulsory secondary education, and other educational levels in formal and nonformal education. The study sample included 115 teachers of technology, programming, and/or robotics (50.4% male, 49.6% female). The research was carried out using a quantitative, retrospective, and cohort methodology through a cross-sectional, non-experimental, and non-longitudinal study over time. A questionnaire specifically designed to collect data from the participating teachers was used. According to the results obtained, educational robotics is a multidisciplinary learning tool that enhances the development of skills such as personal autonomy, collaborative work, and emotional management, motivates the acquisition of knowledge based on practice, promotes curricular sustainability, and creates a new learning context where the teacher is the formative guide of the students and the students are engaged in their own learning.

**Keywords:** pedagogical practices; education; interpersonal skills; intrapersonal skills; robotics; sustainability; sustainable professional development; student engagement

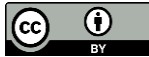

## 1. Introduction

### 1.1. Learning Environment and Sustainable Development Goals

Society is undergoing many scientific, technological, and cultural transformations, and people need to be prepared to face these changes and to lead the commitment to improve the world around us. For this, it is necessary that the population is prepared, and therefore, formative education must be a fundamental premise in this process [1].

Training requires a context adapted to the needs of students, so it is advisable to adapt environments to ensure better student learning. Precisely, the design of learning environments is a strategic planning process for the development of students' skills, favoring comprehension and reflection skills. Technological literacy and skills such as management and/or problem-solving enable and promote the development of competencies that will be useful outside the classroom [2]. For this reason, it is important to highlight formative education based on robotics, adapted to the technological context of today's society.

Educational robotics is a specific learning context, supported by resources such as Information and Communication Technologies (ICT), in which pedagogical mediation processes are required to enable students to acquire competencies related to constructive skills and robot programming through the creative ability and inventiveness of the student [3]. The STEM (Science, Technology, Engineering, and Mathematics) learning discipline and educational robotics are currently highly relevant training areas, although there are few studies linking robotics as a tool for developing educational sustainability [4].

Educational robotics bases its learning methodology on Jean Piaget's constructivism. Within this learning stream, Seymour Papert developed constructivism, a learning context in which the computer is an integral part of student learning during formal education. This learning context, in turn, supports the idea that learning is a process in which the learner plays a major role in achieving their personal and academic development through hands-on experimentation, either through construction and/or manipulation of material resources [5].

The modern teaching process based on robotics implemented in the educational classroom highlights the relevance of educational achievements based on content knowledge and in the progressive development of behavioral, social, scientific, cognitive, and intellectual attitudes and skills of the students [6]. Educational robotics enhances the ability to facilitate and promote the creation of new learning environments, significantly stimulating the teaching and learning process for both students and teachers. Teachers act as guides for student learning in the classroom with the intrinsic objective of developing interpersonal, collaborative, organizational, expository, and/or civic skills [7,8].

Educational robotics is in line with the pedagogical premises [9] of creating motivating learning contexts, assigning a guiding role to the teacher as a facilitator and moderator responsible for managing learning, contributing to the establishment of implicit relationships between different school subjects, facilitating curricular transversality by linking the educational content of the curriculum according to a methodological and strategic teaching process adapted to the needs of the students, and promoting the progressive autonomy of the student body [10]. It is also a favorable [11] formative learning context that can train teachers in their classroom work, providing them with resources and tools that allow them to motivate students in their reflective and critical capacities, stimulating them in the acquisition of scientific knowledge and in the internalization of learning through manipulation and practical interaction.

For all these reasons, educational robotics can be considered a learning domain that implements training activities related to the automation of educational processes as a scripted proposal for the internalization of learning [12].

According to the literature review and case studies conducted by Schina et al. [13], it is relevant to mention how the Sustainable Development Goals (SDGs) can be implemented in the teaching and learning process developed in educational robotics classrooms, such as SDG 3 (health and well-being), SDG 4 (quality education), and SDG 11 (promoting a society based on sustainable education).

*1.2. Intrapersonal and Interpersonal Skills in Educational Robotics Classroom*

Educational robotics is a methodological learning tool that allows the acquisition of research-related skills through the active and continuous work of students, combining the theoretical and practical aspects of learning, analyzing the information collected in order to subsequently substantiate the results obtained and draw conclusions, improving the autonomy and the analytical and reflective capacity of the student [14].

Educational robotics works in areas related to the cognitive, attitudinal, and social development of students, improves effective learning in the classroom, favors alternative learning paths over traditional ones, and enhances the development of educational skills such as logical reasoning, cooperative work, and the automation of practices, and favors personal autonomy [15].

Intrapersonal and interpersonal skills make up a broad learning in identifying and managing emotions. Each has many specific psychological attributes and capabilities. Intrapersonal skills refer to the recognition and/or most appropriate use of one's own emotions, and interpersonal skills are those that are implemented and put into practice to identify, perceive, and/or assimilate the emotions or feelings generated by other people [16]. By implementing robotics in the educational environment, it is possible to develop personal and social skills (interpersonal and intrapersonal) that are essential in active learning, such as critical and reflective thinking, the acquisition of tools and information search processes, the acquisition of guidelines for teamwork, and the development of scientific skills in the context of research, communication, and leadership, thus meeting students' training needs and contributing to the construction of efficient and useful knowledge at a personal, social, and/or academic level [3].

In educational robotics learning sessions, unforeseen events and problems arise, to which the students adapt by generating strategies to solve problems and/or conflicts, combining different perspectives by collaborating in working groups to debate, discuss, and act together to achieve common goals. The interdisciplinary activities based on the STEM learning methodology used in the robotics classroom promote the development of skills at different educational levels and encourage the management and resolution of problems closely linked to sustainability [17].

### 1.3. Students' Motivation in the Educational Robotics Context

In the educational context, students are not always predisposed to do what they should in order to learn, either because they do not have enough willpower to carry out certain tasks or because they are distracted by other daily interests. There are different types of motivation, including intrinsic motivation, which is generated by the activity performed by the student [18], and extrinsic motivation, which is produced by external incentives for the activity performed, whether rewards or punishments, that condition the behavior and interest of the student. This problematic situation, caused by a lack of interest and motivation of students in teaching areas related to science and technology, can be minimized with the methodological implementation proposed through the use of educational robotics, motivating students and utilizing playful learning to encourage university studies and careers related to science and/or engineering [19].

It is advisable for students to focus on tasks in order to achieve long-term goals, which generates satisfaction. Motivation in itself does not produce exceptional or outstanding results from students, but it can create an inherent satisfaction in what they do, creating more energy and strength and encouraging greater interest and desire to achieve their goals [20].

Emotions are essential during teaching and learning (through positive or negative stimulation), so the impact and responsibility of a counselor, teacher, and/or mediator are to manage the emotional capacity of the classroom and to redirect students' interests through strategies and resources to achieve specific goals defined by the school and the teacher [21].

It is important to identify the factors that promote the intrinsic motivation of students in order to be able to guide and advise them in a specific way, adapted to their needs, during the teaching and learning process [22]. Education, in terms of its methodology of social implementation, has evolved relatively little in recent decades. Our ancestors learned with a procedure very similar to the current one (explanation of content, completion of exercises, resolution of doubts, and tasks dedicated to their completion outside school hours), being, therefore, a passive agent in the process of teaching and learning.

### 1.4. Interdisciplinary Curricular Learning, Innovation, and Material Resources

Although current Spanish legislation [23,24] provides for the integration of STEM learning into the curriculum in an interdisciplinary way, the average training of students

in programming and robotics in Spain is very limited, with academic training beginning in secondary education courses at a relatively late age (15–16 years).

According to the study by Román-Graván et al. [25], educational robotics should be implemented from the early stages of primary education. According to Maiz Guijarro and Carvalho [26] in their analysis of the effectiveness of learning educational robotics in early childhood education, it offers a series of remarkable benefits in several specific areas, including cognitive (acquisition and internalization of knowledge), socio-affective (teamwork, internalization and development of social values, consideration, respect, mutual help), and attitudinal (increased motivation, reinforcement of the playful component in the development of activities, creativity).

Today's society demands dynamic and active learning that internalizes, to a great extent, the knowledge acquired in the school context. A practical activity that is effective for learning is one that provides a positive effect or impact on the student that can be quantified in terms of performance, both academic and attitudinal [27]. Pedagogical innovation in the classroom through technological tools favors the teaching practice of teachers, develops critical values and attitudes, favors the active participation of students, and allows an educational context with greater freedom (thinking, feeling, etc.) to interact and actively participate in the classroom.

In accordance with these demands in the educational context, teachers must be trained in various areas of knowledge (mathematics, physics, etc.) in order to adapt to the needs and premises of current technological and multidisciplinary progress based on the development of Computational Thinking (CT) [28]. This type of thinking can be defined as the ability to systematically face problem-solving (using programming, algorithmic coding, etc.), extrapolating the solutions proposed and applying them to other contexts of reality [29].

According to the material resources used in the educational robotics classroom (either free or paid software or hardware in certain institutions), different designs or teaching models can be distinguished for the development of student learning, such as "learning robotics", where students use the robot as an essential tool for the internalization of elementary knowledge of engineering, programming, and/or electronics for the correct practical execution of educational robotics activities; "learning with robotics", a methodological standard where the teacher and students use robots as active support to help in the teaching and learning process; and "learning by robotics" or "robotic-based instruction", a teaching model in which robotics is used as a transversal learning methodology in different subjects or educational curricular areas [30].

The creation and development of activities adapted to the specific learning needs of students are essential in the current educational context. The activities developed in the educational robotics classroom facilitate the internalization of knowledge due to their ability to link the contents and topics studied in the classroom with practical reality, extrapolating to the world around us [31].

Therefore, this research study aims to delve into the characteristic features of robotics and educational programming from the perspective of teachers of technology, robotics, and/or programming as a promoter of sustainability in learning, knowledge, and development of intrapersonal and interpersonal skills of educational curricular sustainability competencies of students.

The research question is: What is the personal and professional perception of teachers of technology, robotics, and/or programming teaching in formal and non-formal academic contexts in relation to the acquisition of social competencies and the development of intrapersonal and interpersonal skills through the use of educational robotics and its potential sustainable curricular development?

In order to answer this question, the following general research objective is defined:

- To investigate the perception of teachers of technology, robotics, and/or programming and to study the social benefits of students' interpersonal and intrapersonal and/or academic skills for improving curriculum sustainability during the teaching–

learning process from the perspective of robotics and programming for students in early childhood education, primary education, compulsory secondary education, and other educational levels in formal and non-formal education.

In relation to this general objective, the following specific objectives are established:

- To review and catalog the academic teaching background, professional experience, and evaluation experience of teachers of technology, programming, and/or educational robotics.
- To know the influence of educational robotics and programming in sustainable learning environments in school teaching and to analyze the material resources used by teachers of technology, programming, and/or robotics during their teaching performance in the educational robotics classroom.
- To study the influence of robotics and/or educational programming on motivation in the teaching and learning process, collaborative work, the development of creativity and imagination, problem-solving, personal autonomy, and emotional management skills of students and skills that can be useful for their academic or professional future.

## 2. Methodology

### 2.1. Methodological Design

The research was conducted using a quantitative, retrospective, cohort methodology through a cross-sectional, non-experimental, non-longitudinal study over time. An online questionnaire specifically designed to collect quantitative and qualitative data from study participants was used.

### 2.2. Sample

The sample involved in the research was selected by means of non-probabilistic and intentional theoretical sampling, which allows the researcher the opportunity to effectively analyze the information [32] according to the criteria, specifications, and demands of the study. The study sample included professionals (professors and teachers of early childhood education, primary education, secondary education, and/or other educational levels) who teach subjects related to technology, robotics, and/or programming in formal and non-formal education environments (education at curricular and extracurricular levels), both in public educational institutions and in subsidized and/or private centers.

The study sample, described in Table 1, included 115 participants, of whom 49.6% were women and 50.4% were men, with an average age of 35.7 years. At the early childhood education stage, the proportion of female teachers is significantly higher than that of male teachers, registering 83.3% compared to 16.7%. This trend changes markedly in primary education, where the presence of male teachers, at 50.8%, is slightly higher than that of female teachers, at 49.2%. In compulsory secondary education, the presence of male teachers is again higher, with 56.7% compared to 43.3% of female teachers. The same occurs at other educational levels, such as Baccalaureate, vocational training, and/or university, where 66.7% are men and 33.3% are women.

**Table 1.** Distribution of the total sample according to the educational level of teaching and the gender of the participating teachers.

| Educational Level | Men | | | Women | | | Total | |
|---|---|---|---|---|---|---|---|---|
| | N | % of Level | % of Total | N | % of Level | % of Total | N | % |
| Early Childhood Education | 2 | 16.7 | 1.7% | 10 | 83.3% | 8.7% | 12 | 10.4% |
| Primary Education | 31 | 50.8% | 27% | 30 | 49.2% | 26.1% | 61 | 53% |
| Secondary Education | 17 | 56.7% | 14.8% | 13 | 43.3% | 11.3% | 30 | 26.1% |
| Other Educational Levels | 8 | 66.7% | 7% | 4 | 33.3% | 3.5% | 12 | 10.4% |
| Total | 58 | - | 50.4% | 57 | - | 49.6% | 115 | 100% |

The employment status of the teachers is classified according to their participation and/or involvement in educational activities in the non-formal (43.5%) and formal (56.5%) spheres. Table 2 shows the employment/professional status of the teachers participating in the study.

**Table 2.** Employment/professional status of technology, programming, and/or robotics teachers.

| Teacher's Employment/Administrative Status | N | Percentage (%) |
|---|---|---|
| Teaching staff for extracurricular activities | 50 | 43.5% |
| Private/subsidized school teachers | 44 | 38.3% |
| Interim public school personnel | 12 | 10.4% |
| Public school staff | 8 | 7% |
| University teaching and research staff | 1 | 0.9% |
| Total | 115 | 100% |

Table 3 below shows the age ranges to which the teachers belong.

**Table 3.** Age ranges to which teachers of technology, programming, and/or educational robotics belong.

| Teacher Age Range | N | Percentage (%) |
|---|---|---|
| Between 18 and 25 years old | 9 | 7.9% |
| Between 26 and 34 years old | 58 | 50.4% |
| Between 35 and 45 years old | 29 | 25.2% |
| Over 45 years old | 19 | 16.5% |
| Total | 115 | 100% |

The participating teachers work in various autonomous communities in Spain (94.7% of the sample) and internationally (5.3%), as shown in Table 4.

**Table 4.** Distribution of educational robotics teachers with respect to the autonomous community where they carry out their teaching and/or professional activity in the Spanish national and international territory.

| Spanish National Level | | |
|---|---|---|
| **Autonomous Community** | **Frequency** | |
| | **N** | **Percentage (%)** |
| Andalusia | 13 | 11.3% |
| Aragon | 2 | 1.7% |
| Asturias | 8 | 7% |
| Cantabria | 2 | 1.7% |
| Catalonia | 23 | 20% |
| Castilla y Leon | 6 | 5.2% |
| Valencian Community | 6 | 5.2% |
| Extremadura | 4 | 3.5% |
| Galicia | 6 | 5.2% |
| Balearic Islands | 1 | 0.9% |
| Canary Islands | 2 | 1.7% |
| La Rioja | 1 | 0.9% |
| Madrid | 29 | 25.2% |
| Murcia | 3 | 2.6% |
| Pais Vasco | 3 | 2.6% |
| International Level | | |

| Country | N | Percentage (%) |
|---|---|---|
| Argentina | 4 | 3.5% |
| Peru | 1 | 0.9% |
| United States | 1 | 0.9% |
| Total | 115 | 100% |

*2.3. Instruments*

The research was conducted using a questionnaire created with the online tool Google Forms.

Teacher Questionnaire

Data collection from robotics teachers was performed using a questionnaire developed and validated by Cabello Ochoa and Carrera Farran [33]. This instrument aims to explore teachers' attitudes and beliefs regarding robotics and programming in the educational setting. The questionnaire consists of 35 quantitative items, including Likert-type and multiple-choice questions, as well as 2 qualitative items in the form of short, semi-open-ended questions to complement the previously collected information.

The questionnaire is made up of 37 items, which are grouped into two sets of items.

—Set 1 (8 items): Teacher profile. It contains questions that address aspects related to the teacher's professional profile, formulated on the basis of questions 1–8.

—Set 2 (29 items): Teacher knowledge and involvement in educational robotics. Includes questions on teacher familiarity and involvement in robotics teaching, addressed in items 9–37.

*2.4. Procedure*

Data were collected with a questionnaire on several contact platforms, including the institutional email of the University of Burgos and professional or work networks.

Teachers completed the questionnaire online, answering the questions in an average time of 15–20 min. Anonymity and confidentiality with respect to the personal and professional identity of the respondents were guaranteed.

Data collection was conducted during the 2022–2023 and 2023–2024 academic years.

*2.5. Analysis*

After collecting the data from the teachers' survey, the quantitative information was analyzed by means of a frequency analysis with descriptive statistics of the sample and cross tables, considering relevant variables and questions. Descriptive statistical analyses were performed on the quantitative data related to the variables studied. In turn, cross-tables were created to compare data between different relevant variables under study.

With regard to the qualitative data, a descriptive and interpretative analysis was carried out to assess and interpret the experiential information collected. For the categorization of the information, a first level axial and open coding was applied, labeling and structuring the contents and variables of the study.

For the representation and descriptive analysis of the resulting values, various tools/software were used, including the statistical program IBM SPSS Statistics v.25 and the Microsoft Excel 2019 calculation processor. These programs facilitated the visual presentation and detailed analysis of the results obtained in the research.

**3. Results**

The data collected in the research study were classified according to their corresponding variables and the content associated with them. This organization is detailed in Table 5, which provides the representation and arrangement of the results obtained.

**Table 5.** Classification of the variables and contents treated in the research study, results and arrangement of items according to the information of the instrument used in the research by Cabello Ochoa and Carrera Farran [33].

| Categories, Study Variables, and Contents | | | Item |
|---|---|---|---|
| | **Categories** | **Contents** | **No.** |
| Block 1. Teacher professional profile | Personal and academic profile of the teacher | Age | 1 |
| | | Genre | 2 |
| | | Academic qualification | 3 |
| | Professional profile of the teacher | Time of teaching experience | 4 |
| | | Autonomous community where teaching activity is carried out | 5 |
| | | Administrative status | 6 |
| | Teacher's professional activity | Level of education at which the professional activity is carried out | 7 |
| | | Curricular areas in which most of their professional activity is developed | 8 |
| | | Type of material used during the teaching activity | 13 |
| | | Evaluation of teaching experience | 16 |
| Block 2. Teacher knowledge and involvement in educational robotics | Formative skills developed in educational robotics. | Intrapersonal Skills — Creativity development | 15.6 |
| | | Intrapersonal Skills — Development of emotion management skills | 15.7 |
| | | Intrapersonal Skills — Development of personal autonomy | 15.9 |
| | | Interpersonal Skills — Motivation development | 15.10 |
| | | Interpersonal Skills — Development of personal/academic/professional skills for the future | 17 |
| | | Interpersonal Skills — Development of collaborative work | 17 |

*3.1. Results According to Academic Background, Professional Experience, Assessment of Teaching Experience, and Material Used in Educational Robotics Classroom*

3.1.1. Academic Training, Professional Experience, and Assessment of Teaching Experience

The results of academic teaching background, professional experience, and evaluation experience of teachers of technology, programming and/or educational robotics are indicated below.

According to the 'academic background of the professionals'(item 3), 59.1% of the teachers who teach technology, programming, and/or robotics have an academic degree related to the field of Education, followed by Engineering and Architecture (18.3%), Social and Legal Sciences (4.3%), and Health Sciences (2.6%), as the most representative.

The participants in the research study have 'teaching experience' (item 4) of less than 5 years in 40.9% of the cases, 5 to 10 years in 36.5% of the cases, 11 to 15 years in 12.2% of the cases, and more than 15 years in 10.4% of the cases.

The teaching team evaluates their 'experience as a teacher'(item 16) or professor of educational robotics with an average score of 4.44 out of 5 on a scale, where the values have been set from 1 (very disappointing) to 5 (very enriching), reflecting a generally positive and enriching perception of the educational robotics teaching experience.

3.1.2. Materials Used by the Teacher in the Robotics Classroom

To know the influence of educational robotics and programming in sustainable learning environments in school teaching, it is important to study the material resources used by teachers of technology, programming, and/or robotics in their educational robotics classrooms.

The types of material used most by teachers during the course of the teaching activity (item 13) are LEGO WeDo 2 (62.6%), 3D printers (58.3%), LEGO WeDo (52.2%), LEGO MINDSTORMS EV3 (51.3%), virtual reality using 3D glasses (28.7%), LEGO SPIKE Prime (22.6%), Sphero Balls (20.9%), programming using the electronic boards Arduino(16.5%) and Scratch(14.8%), 3D pens (12.2%), the programming bee Bee-Bot in early childhood education (11.3%), Makey Makey (12.1%), and 3D design using software such as Tinkercad (free online version) and SketchUp (free version) (5.2%).

The material used by students in the robotics classroom motivates them to design and configure their own creations, allowing them to be the main actors in the learning process and not a mere consumer, encouraging the use of materials and instruments that are part of their daily lives, promoting sustainable learning (item 17).

—Teacher 70: "Robotics material encourages teaching in which students are not just consumers, but actors and main creators of their own learning material".

—Teacher 17: "Robotics increases students' motivation by teaching them how to use material resources that are part of their everyday life".

The materials used are very diverse and can be adapted taking into account the economic resources of the school and/or institution that uses them in different curricular areas (item 17). At the same time, mention is made of the culture of recycling and sustainability during the use of the technological material used to carry out the activities in the educational robotics classroom.

—Teacher 22: "As a teacher I usually do a project with recycled robots: we build "a robot" (without a programming board) where the students think and answer the questions, cut out the cardboard to make the robot and then connect it to a programming board, see how the motors work and behave…".

—Teacher 100: "Despite being a discipline that involves the use of digital technologies, the manipulation and creation of objects (sensors, actuators or mechanisms) can be done in a traditional way (DIY or MAKER). This allows to promote the culture of recycling and sustainability. In addition, small modifications to a project give rise to a wide variety of new projects. All it takes is a little imagination".

*3.2. Knowledge Results and Teaching Involvement*

The following is the study of the influence of robotics and/or educational programming in the teaching and learning process in the development of intrapersonal and interpersonal skills.

3.2.1. Intrapersonal Skills and Training Competencies Developed using Educational Robotics

Educational robotics enhances the development of 'creativity' (item 15.6) in students according to the teachers' perception (totally agree: 68.7%; quite agree: 29.6%; slightly agree: 1.7%). According to the qualitative responses of the teachers, the following perspectives are presented (item 17).

Creativity allows students to achieve greater solvency, autonomy, and development of imagination in solving questions and problems, favoring developing their own proposals and ideas, being aware of the tools they have at their disposal and using them appropriately and in context and to break down barriers of different kinds, to provide originality and inventiveness, and to be protagonists in their learning as expressed by the teachers.

—Teacher 12: "Students develop their creativity in the classroom, which generates the expectation that they will surpass themselves with proposals to solve their own challenges".

—Teacher 13: "Robotics invites us to develop creativity, because there are different ways to achieve the same result".

—Teacher 35: "Working with tangible tools such as robots and constructions made with your own hands fosters creativity and allows you to put into practice a wide range of theoretical knowledge from any academic discipline".

—Teacher 104: "Learning through educational robotics in the classroom facilitates and encourages students to be creative in programming and using their robot in a variety of ways".

—Teacher 109: "Robotics encourage critical thinking and stimulates creativity".

Educational robotics favors the 'personal autonomy' (item 15.9) of students according to the teachers' opinions (totally agree: 43.5%; quite agree: 48.7%; slightly agree: 7.8%), and as it is well analyzed in their qualitative answers (item 17), robotics as a learning discipline helps to develop thinking and skills such as fine psychomotor skills, improving the manipulation of objects.

—Teacher 2: "Robotics increases student confidence, which impacts all areas of learning, as well as soft skills".

—Teacher 13: "Robotics has the ability to help students become more critical and independent people".

—Teacher 50: "The benefits of robotics stand out in aspects such as fine motor skills in early childhood education students".

—Teacher 75: "Students become autonomous and responsible for their decisions through trial and error".

Teacher 91: "Enhances the autonomy of students in projects that are progressively more demanding in terms of difficulty, solving problems that require greater technological knowledge".

Linked to the concept of "autonomy", several teachers state that educational robotics can develop skills to improve students' emotional management, allowing students to control their emotions in difficult situations when faced with a problem, not give up, have patience and persevere and, thus, develop emotional sustainability in the educational context.

—Teacher 13: "Robotics allows them to develop different skills that are necessary and little exploited in the current educational system: it helps them to have patience, not to give up and to persevere".

Emotional management can also be observed in relation to frustration tolerance in the face of unexpected problems that may arise during activities and the development of logical reasoning.

—Teacher 68: "Robotics in educational classroom develops skills and competencies related to learning how to deal with failure and frustration when things don't work out the first time".

Educational robotics, according to the teachers, helps to improve self-critical and divergent thinking, increasing personal confidence, self-esteem, independence, and being more responsible for their opinions and decisions.

—Teacher 11: "Robotics promote the development of reasoning and logical thinking to a greater extent".

—Teacher 26: "Educational robotics offers valuable formative benefits by promoting critical thinking".

—Teacher 58: "A parameter that I consider is indispensable in the educational classroom is the stimulation of thinking, not only critical, but also self-critical and divergent".

—Teacher 83: "Robotics promotes structured thinking, logical thinking and helps to carry out projects through planning phases, following an established didactic design, but with some freedom".

Educational robotics increases 'students' motivation' (item 15.10) during learning according to teachers (totally agree: 55.7%; quite agree: 38.3%; slightly agree: 6.1%). In item 17, teachers state that robotics motivates and stimulates students to learn, increasing their interest and curiosity to continue learning, with a consequent improvement in the classroom.

—Teacher 9: "The most important thing for me in teaching is the motivational part of the students; to see what they are able to achieve with robotics, encourages them to continue learning, and most of the time, from a leisure and fun perspective".

—Teacher 13: "Robotics is an area of learning that "breaks" the mold: it favors greater interaction among students, motivates group work and allows teachers greater freedom in the process of guiding students in their learning in a less exhaustive manner with respect to the syllabus or content of the subject and with greater methodological freedom".

—Teacher 35: "The motivational part is something to take into account when we talk about learning in educational robotics environments".

—Teacher 85: "Robotics encourages curiosity, interest, motivation and facilitates new learning".

The importance of the teacher and his or her teaching methodology as a fundamental premise for improving such motivation, as a facilitator and guidance of new learning with greater freedom in the interdisciplinary teaching process is revealed.

—Teacher 53: "Educational robotics, as a learning tool, makes sense when students know enough (content) to be able to carry out their own project. In this scenario, not only is there a practical application of the technological process, but also, since it is their own project, the involvement is greater, and so is the intrinsic motivation during the performance of the activity".

—Teacher 90: "Robotics in itself is a very motivating element that can be worked on in an interdisciplinary way in different subjects or educational areas".

—Teacher 96: "We start from the premise that students are very motivated by robotics and programming, which is a very important starting point to work on any curricular content".

According to the responses of the teachers participating in the study, Table 6 synthesizes the most relevant information regarding the intrapersonal skills developed in the educational robotics classroom.

**Table 6.** The most relevant information collected from the sample of technology, programming and/or robotics teachers on the intrapersonal skills developed in the educational robotics classroom.

| Intrapersonal Skills and Training Competencies Developed in Educational Robotics. | | | |
|---|---|---|---|
| **Creativity** | **Personal Autonomy** | **Emotional Management** | **Motivation** |
| There are different ways to achieve the same result. | Development of reasoning and logical thinking to a greater extent. | Develop different skills: it helps them to have patience, not to give up, and to persevere. | See what they are able to achieve with robotics, encourages them to continue learning. |
| Surpass themselves with proposals to solve their own challenges. | Benefits fine motor skills in early childhood education students. | | Greater interaction among students; motivates group work and allows teachers greater freedom in the process of guiding students in their learning. |
| Working with tangible tools such as robots and constructions made with your own hands fosters creativity. | Stimulation of thinking, not only critical, but also self-critical and divergent. | Competencies related to learning how to deal with failure and frustration when things do not work out the first time. | Encourages curiosity, interest, and motivation and facilitates new learning. |
| | It helps improve learning based on autonomy and a better understanding of the real world. | | Can be worked on in an interdisciplinary way in different subjects or educational areas. |
| Learning through educational robotics in the classroom facilitates and encourages students to be creative in programming. | Students become autonomous and responsible for their decisions through trial and error. | | |

### 3.2.2. Interpersonal Skills and Training Competencies Developed in Educational Robotics

Educational robotics facilitates 'collaborative work' (item 15.7) for the teaching team (totally agree: 55.7%; quite agree: 35.7%; slightly agree: 7.8%; do not agree: 0.9%), as can

be seen in the following answers to the questions in item 17, where teachers state how collaborative work allows working on social and interpersonal skills and organization of tasks, as well as active communication and critical thinking (sharing responsibilities, debating opinions and/or perspectives with colleagues), where it is sought that everyone contributes ideas in an equitable way in the development of team projects.

—Teacher 43: "Educational robotics develops critical thinking during teamwork (…) Students are able to distinguish the different profiles or roles at the time of work (the one who builds, programs, organizes, communicates…)".

—Teacher 58: "When I organize my students in groups to build robots, to prepare a program, to create a construction with Lego materials, etc., I see that they learn to work together, to communicate and, above all, to share responsibilities (with each other), I see that they learn to collaborate, to communicate and, especially, to share mutual responsibilities (together). I try to strengthen their social skills and their sense of community".

—Teacher 73: "Educational robotics helps in the development of cooperative learning, inclusion and active participation…".

Collaborative work in the robotics classroom is linked to the improvement of student's motivation, feeling involved during the learning process, and acquiring shared values to promote a better, more inclusive, and sustainable world in which companionship and diversity are basic and fundamental pillars.

—Teacher 26: "In my case, I have implemented educational robotics activities outside of regular education, with people with functional diversity, and it helps me to improve attention and interest, as well as in the acquisition of other knowledge by the students".

—Teacher 58: "It is gratifying to see how, by combining a participatory and collaborative "methodological environment", students are more motivated to want to work (…) if we want to work and promote a future society in which values are shared with the aim of promoting a better world, we must stimulate them and make them see that teamwork and collaboration are essential".

—Teacher 68: "Robotics encourages the inclusion of girls in the field of technology…".

Educational robotics develops 'skills that can be useful for the future' (item 17). Teachers ensure that robotics students are better prepared to face and adapt to the sustainable and technological digital world that shapes society, providing them with a greater number of opportunities and a set of specific tools to face future challenges, both academically and in the workplace and/or career.

—Teacher 27: "Robotics prepares students to face the challenges of today's world and future career opportunities".

—Teacher 30: "Robotics favors teaching in accordance with the educational and professional needs that students will need for current and future professions (...) fostering the development of competencies and skills necessary to provide adequate answers with the logical use of technology in the classroom".

—Teacher 43: "Robotics in an educational environment develops capacities or skills to be able to face the jobs and technologies that may arise in the future. Working with programming and robotics in school will enable students to choose to study careers related to new technologies in the future".

—Teacher 79: "Robotics is an important area of learning that provides students with tools that will allow them to enter the world of work".

—Teacher 96: "The future job opportunities offered by robotics at the educational level are very broad, so it is a very useful investment for students".

Moreover, educational robotics enhances the development of technological skills and competencies, improving time management and student productivity for today and future society.

—Teacher 100: "In the near future, digital technologies will be present in most everyday activities. Automation will make it possible to better manage time and productivity. Therefore, it will be essential to understand how technological processes work in order to

apply for many jobs. Without an initial grounding, many people will be lost in a fully automated situation (as is now the case with older people)".

—Teacher 109: "Educational robotics is a field of learning that prepares for the future, in the sense that it introduces concepts that are needed as a basis for the conception of today's society".

According to the responses of the teachers participating in the study, Table 7 synthesizes the most relevant information regarding the interpersonal skills developed in the educational robotics classroom.

**Table 7.** The most relevant information collected from the sample of technology, programming and/or robotics teachers on the interpersonal skills developed in the educational robotics classroom.

| Interpersonal Skills and Training Competencies Developed in Educational Robotics. | |
| --- | --- |
| **Collaborative Work** | **Skills That Can Be Useful for the Future** |
| Collaboration and dynamic teamwork, since everyone is expected to contribute and help each other when developing joint projects. | Prepares students to face the challenges of today's world and future career opportunities. |
| With functional diversity, collaborative work helps improve attention and interest, as well as the acquisition of other knowledge by the students, and develops critical thinking during teamwork. | Favors teaching in accordance with educational and professional needs. |
| By combining a participatory and collaborative "methodological environment", students are more motivated to want to work. | Effective development of competencies in line with today's world, especially technological and digital competencies. |
| I see that they learn to work together, to communicate, and, above all, to share responsibilities (with each other). | |
| Robotics encourages the inclusion of girls in the field of technology and helps in the development of cooperative learning, inclusion, and active participation. | Automation will make it possible to manage time and productivity better, and it is essential to understand how technological processes work in order to apply for many jobs. |

## 4. Discussion

According to the results obtained from the perception of teachers of technology, robotics, and/or programming, with regard to the concept of educational robotics as a teaching–learning methodology in the academic formative development for improving curricular sustainability and the development of intrapersonal and interpersonal skills of students in early childhood education, primary education, compulsory secondary education, and other educational levels in the formal and non-formal educational environments, the following initial discussions and conclusions are presented.

### 4.1. Academic Background, Professional Experience, Assessment of Teaching Experience, and Material Used in Educational Robotics Classroom

The results of the academic teacher education, professional experience, the evaluative experience of teachers of technology, programming, and/or educational robotics, and the material and didactic guides used in robotics classrooms are discussed below.

The majority of the technology, programming, and/or robotics teachers who carry out their professional work in the classroom have a formal education and/or academic degree related to the field of education, engineering and architecture, social and legal sciences, and health sciences. Teachers value their professional experience in the area of educational robotics as very positive and enriching.

The implementation of didactic guides and the incorporation of the respective material resources and/or tools used in the educational robotics classroom lead to the creation of new learning environments that stimulate and inspire students to explore and experiment with educational technology, learning from it and through it [34].

For the proper development of the teaching exercise in the educational robotics classroom, the teaching team must have at its disposal various material resources and activities specifically designed for implementation in the educational classroom. Among the materials used more frequently by educational robotics teachers in the classroom are LEGO WeDo, LEGO WeDo 2, LEGO MINDSTORMS EV3, and Lego SPIKE Prime, through freely available licensed software, all of which are material resources designed for the purpose of building educational robots and programming their movements using motors and sensors that delimit the way they operate, including pulse sensors, ultrasonic sensors, color sensors, and motion sensors/gyroscopes.

Other materials used by teachers in educational robotics classrooms are Tinkercad (free online version) and SketchUp (free version) software for designing a wide variety of three-dimensional figures to be printed on a 3D printer, virtual reality glasses, Sphero Balls programmable spheres, Arduino (free online version) and Scratch software (3.xx free version) for programming computer language, 3D pens to create three-dimensional figures, programming the Bee-Bot bee, which favors memorization and spatial vision for students in early childhood education in their first formative years, and electronic boards such as Makey Makey, which allows students to discover how electronic circuits work and the ability of different materials to conduct electricity.

The manipulation and practical interaction with these specific robotics materials improve the performance of executive operations and task planning with the aim of solving certain situations that arise in the classroom through decision-making, the implementation of computational thinking, and the fragmentation of large and complex problems into others that are more concrete, simple, and easy to face and solve. Activities designed on the basis of educational robotics contribute to improvements in problem management, promoting student reflection while the teacher acts as a guide and assists during the realization of activities linked to the real context [2,8]. From the perspective of the educational robotics teaching team, students can be involved in achieving the principles of sustainable design, as the participating teachers agree with the usefulness and efficiency of the practical activities in which educational robots are used.

It is important to emphasize that the use of diverse educational resources in the robotics classroom makes it possible to adjust the economic cost of the materials and tools to the possibilities of the educational centers, with low-cost possibilities for an economical didactic implementation [35]. This allows the implementation of practical sustainability during the teaching process in the school classroom, being able to reuse computer and electronic materials for other purposes and adapt them to activities oriented to the learning of educational robotics (programming, 3D design, etc.).

### 4.2. Learning Environment, Intrapersonal, Interpersonal Skills, and Training Competencies Developed in Educational Robotics

In the following, the influence of robotics and/or educational programming in the teaching and learning process on the development of intrapersonal and interpersonal competencies is discussed.

According to the results obtained, educational robotics acts as a multidisciplinary learning tool that enhances the development of skills such as personal autonomy, collaborative work, and emotion management, motivates the acquisition of knowledge based on practice, promotes curricular sustainability, motivates creativity, and creates a new learning context where the teacher is the formative guide of the students and the students are the protagonists of their own learning. According to the research conducted by Chatzopoulos et al. [4], education based on the interdisciplinary teaching discipline STEM fosters and promotes positive learning attitudes aimed at cultural and social

sustainability. In robotics workshops, students develop a predisposition to developing skills such as teamwork, stress management and control, problem-solving, improved self-esteem and assertive decision-making, and the ability to think critically and on the basis of emotion management [36].

Robotics favors work motivation, allowing a deeper integration of different areas of knowledge for problem-solving [37,38]. Educational robotics actively contributes to the better development of teamwork in the classroom, learning to respect the opinion of others, and joining efforts and reaching a common consensus to achieve a more homogeneous solution among all team members. The significant learning that takes place through experimentation encourages strategic learning, which increases the motivation and participation of the students, who try to connect the experiences in the activities developed in the classroom with the reality of the world around them [39].

The research study conducted by Chin et al. [40] regarding the impact that educational robots have on the learning of primary school students included a control group of students learning through the traditional methodology of exposure or PowerPoint presentations and an experimental group with presentation and exposure combined with the use of educational robots. It concluded that robotics implemented in the classroom improves the learning performance of students. These improvements in learning performance are attributed to increased motivation on the part of the students during the educational process, which improves their focused attention during the teacher's explanations, increases their interest in the subject matter, and increases their confidence to participate in the proposed activities, providing new stimuli and feedback.

Robotics is closely linked to cooperative learning in the implementation of educational projects [12], stimulating students to work and strengthening social skills and cooperation to achieve a common goal. Collaborative work must, therefore, take place in a context where respect for the diversity of people, opinions, and perspectives is a fundamental requirement for sustainable education. For all the above reasons, creativity and teamwork are benefits of robotics to minimize discrimination and improve the classroom environment and student learning to avoid school failure [41]. For his part, Ref. [42] considers that the design and assembly of robots are part of collaborative learning that promotes creative stimulation in addressing and solving various problems that arise during the development of activities, awakening and stimulating the curiosity of students. For this purpose, in educational robotics classrooms, the robots built by the students comply with the principles of sustainable design through collaboration in working groups and based on learning based on the pedagogical process of teaching in the educational classroom [4].

The usefulness of the knowledge acquired in robotics allows the development of useful skills for the future, allowing students to better adapt to the digital, sustainable, and technological world and giving them more opportunities to face future academic and labor/professional challenges. Robotics enables the development of thinking skills from a very early age (3–6 years) in order to demonstrate the significant learning of this discipline for the development of technological skills and future academic challenges [43].

Strengthening and improving students' personal autonomy is a relevant aspect for teachers. Patience and perseverance promote resilience, initiative, critical and divergent thinking, and tolerance to frustration in the face of problems that may arise during the development of activities. Precisely, during multidisciplinary learning with technologies (STEAM), among other skills, autonomy and entrepreneurship are developed [22].

Inquiry and research to find solutions to the problems posed in robotics activities and linked to real life enhances the implementation of cooperative work in a variety of different contexts and among students with unique and diverse skills and abilities [44].

Students, therefore, need to be prepared to manage and cope with situations and problems, so improving frustration control is one of the aspects where robotics can help them to improve their academic skills and sustainable emotional learning continued over time (emotional management). Critical thinking (responsibility for the activities and projects that are carried out), sociability, tolerance, and the student's permanent search for

self-realization are some of the skills that are promoted and developed in the educational robotics classroom [45].

In accordance with all this and to conclude, it is advisable that students begin their sustainability robotics education at an early age, through sustained and continuous learning over time, taking into account the development of social, emotional, intrapersonal, and interpersonal skills as elementary support to generate competencies such as leadership [1], generating an improvement in critical and reflective thinking through study, analysis, and problem-solving.

## 5. Conclusions

Educational robotics is a learning methodology that enhances the mastery of interpersonal and intrapersonal technological skills and the competencies of students. It boosts students' creativity, imagination, and inventiveness through technological tools used to create, design, and build their own creations. Likewise, hands-on manipulation and interaction is a component that improves the students' executive operations, promoting several sustainable development goals, mainly that of sustainable education, health and well-being (SDG 3), quality education (SDG 4), and promoting a society based on sustainable education (SDG 11).

For this purpose, the materials used in the robotics classroom are very diverse and can be adapted based on the economic resources of the school and/or institution that uses them in different curricular areas, which ensures economic sustainability and makes the associated costs of the material resources profitable in agreement with the teachers.

Creativity and the playful component of learning contribute to a key aspect of the educational classroom: motivation. To achieve this, it is essential to have material resources and their respective didactic units and activities specifically designed for the educational robotics class.

The motivational component of educational robotics makes the teaching and learning process more efficient during the formative development of the student. There are two essential factors that determine the quality of the learning obtained: the teacher and their method of implementation in the classroom. Motivation increases the student's interest in wanting to know about a specific subject, to understand how it works, and to put into practice something they did not know before.

The possibilities generated in the robotics classroom to solve possible problematic situations raise a series of personal and emotional stimulations for the student, giving them greater satisfaction and a formative path that encourages them to pursue challenges and achieve sustainable goals in their continuing education. In fact, the challenges that students have to face openly expose emotions and complex situations that need to be managed in the context in which they develop, either individually or in work groups.

The teacher in the educational robotics classroom is an indispensable element in the training process, being a guide or reference that manages the factors that determine the good environment generated in the classroom, orienting the parameters and steps to be followed through sustainable learning and stimulating the student to perform activities independently with greater solvency.

The development of the student's autonomy in managing and solving diverse situations and/or complex problems is essential for the teacher in their daily management in the classroom, which is why it is so important in theoretical and practical teaching. Developing, strengthening, and improving patience, perseverance, the initiative to innovate, and frustration tolerance are also factors that the teacher of technology, robotics, and/or programming has as a main reference in their work in the classroom of educational robotics, being fundamental premises for the development of sustainable educational teaching.

The activities developed in the classroom have an additional learning component: collaborative work. For the teacher, robotics is an educational area that encourages collaborative work by students in order to manage and solve issues of various kinds in working groups and to overcome problems in a context similar to reality, contextualized according

to the social and cultural sustainability that is generated in the society that surrounds us and in which we live.

Learning based on the development of emotional skills improves the acquisition of social skills. The inclusion of students in the educational robotics classroom, therefore, favors the emotional sustainability of the students, allowing a coexistence based on equal rights, assertive communication, and respect during the teaching and learning process.

Acquiring knowledge in robotics is valuable as it fosters the development of practical skills essential for the future. This empowers students to adeptly navigate the digital, sustainable, and technological landscape, equipping them with enhanced opportunities to tackle academic and professional challenges ahead. From as early as 3 to 6 years old, robotics nurtures critical thinking abilities, underscoring its importance in cultivating technological proficiency and preparing for their academic and/or professional future.

In conclusion, the development of interpersonal and intrapersonal skills has a nexus of union during the teaching and learning process based on robotics and/or educational programming. This compendium of skills developed by the teacher and the students in the educational robotics classroom promotes curricular sustainability and the development of effective competencies and encourages the possible usefulness and personal, academic, and/or professional future of the student in multiple contexts.

Despite the results and conclusions obtained, this study is not without some possible limitations.

The results obtained are based on a single type of instrument, the questionnaire. It is important to highlight that collecting information from teachers through qualitative instruments such as individualized or group interviews (focus groups) could provide a wider range of nuances in the perception of the participants, thus enriching the analysis and understanding of the teaching experience in the field of educational robotics and complementing the data for future research.

The study sample is composed of 115 teachers of technology, programming, and/or robotics, which is an optimal number to collect data and information but could be larger in subsequent studies.

Finally, obtaining information regarding the perception of students and families [46] could complement the perceptions provided by teachers, with the purpose of helping future research design didactic units and educational projects according to the underlying needs of students during their learning process and to develop and/or improve teaching methodologies.

**Author Contributions:** Conceptualization, V.A.-V.; Methodology, D.L.-R. and V.A.-G.; Formal analysis, D.L.-R. and V.A.-G.; Resources, V.A.-V.; Writing—original draft, D.L.-R.; Writing—review & editing, V.A.-V. and V.A.-G. All authors have read and agreed to the published version of the manuscript.

**Funding:** This research received no external funding. It is part of the Doctoral Thesis written by David Llanos-Ruiz and directed by Vanesa Ausin-Villaverde and Victor Abella-Garcia.

**Institutional Review Board Statement:** The study was conducted according to the guidelines of the Declaration of Helsinki and approved by the Ethics Committee of University of Burgos (IO 31/2024; date of approval: 9 May 2024).

**Informed Consent Statement:** Informed consent was obtained from all participants involved in the study.

**Data Availability Statement:** Data is contained within the article.

**Acknowledgments:** The authors would like to acknowledge and express their gratitude to all the teachers who participated and collaborated to collect the necessary information to make this research possible.

**Conflicts of Interest:** The authors declare no conflicts of interest.

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
