# Peer review of "Interpersonal and Intrapersonal Skills for Sustainability in the Educational Robotics Classroom"

_sustainability, doi:10.3390/su16114503_

Round 1

Reviewer 1 Report

Comments and Suggestions for Authors

The article "Interpersonal and Intrapersonal Skills for Sustainability in the Educational Robotics Classroom" concerns an important issue of robotic education at school, its impact on the development of students 'competences and teachers' approach to this issue.

The article was presented in an interesting way and I am glad that I could read it.

Most of the cited literature comes from recent years.

Before his final publication, however, it is worth paying attention to several issues:

1) introduction seems a bit too long, I encourage you to think about the separation of, for example, a chapter on the issue of robotics and its impact on education,

2) it would be good to present the response structure with more important questions on the charts to improve the readability of the article and display the most important results,

3) it is also worth considering the display of individual blocks of content - e.g. in chapter 3.2.1 all considerations merge into one whole. Maybe quoting individual teachers on the basis of a point list would help to some extent?

4) lines 187-193 - here a point list was used, but there is only 1 research objective here. Is it justified in that case?

5) in terms of research objective - is “the analysis” research object in itself? The analysis should rather lead us to the goal, and not be an end in itself,

6) after table titles (e.g. Table 5) there are 2 dots for some reason,

7) line 299 - after the number "59.1%" there are several unnecessary spaces. This situation also appears in other places (e.g. line 154).

I encourage you to think about the above issues - their implementation may, in my opinion, improve the quality of the article and its readability.

Author Response

REVIEWER 1 

The changes introduced are indicated in gray color for better visualization: 

1) Introduction seems a bit too long, I encourage you to think about the separation of, for example, a chapter on the issue of robotics and its impact on education. 

The introduction has been reorganized and subdivided into four more specific and concise sections to make it easier to read and to avoid it being too long. More recent articles have also been added to the “references” to highlight their relevance to modern teaching practices as stated by another reviewer (highlighted in gray in the review document):

1.1. Learning environments and Sustainable Development Goals 

1.2. Intrapersonal and interpersonal Skills in educational robotics classroom 

1.3. Students' motivation in educational robotics context 

1.4. Interdisciplinary curricular learning, innovation and material resources 

(Lines 34-193). 

2) It would be good to present the response structure with more important questions on the charts to improve the readability of the article and display the most important results. 

The results of the research have been restructured, organizing the information in a more concrete way in each of the aspects dealt with, subdividing it into two more concise sections for better visualization (3.2.1. Intrapersonal Skills and training competencies developed in educational robotics and 3.2.2.) Each of the items in these sections (creativity, personal autonomy, collaborative work...) are structured in an individualized way, showing detailed results and then the teachers' answers in each of them. Complementing this, two tables summarizing and highlighting the key contributions (most relevant teacher responses) of each of these sub-sections are presented in tables 6 and 7).  (Lines 314-547).

Added two tables to synthesize the most important responses of the teachers in the study in each of the intrapersonal and interpersonal skills:  

  • Table 6 (line 473). Most relevant information collected from the sample of technology, programming and/or robotics teachers on the intrapersonal skills developed in the educational robotics classroom 
  • Table 7 (line 544). Most relevant information collected from the sample of technology, programming and/or robotics teachers on the interpersonal skills developed in the educational robotics classroom. 

3) it is also worth considering the display of individual blocks of content - e.g. in chapter 3.2.1 all considerations merge into one whole. Maybe quoting individual teachers on the basis of a point list would help to some extent? 

As mentioned above, it is advisable to differentiate part of the results provided in point 3.2. Knowledge results and teaching involvement, so this section has been subdivided into two separate sections: 

3.2.1. Intrapersonal Skills and training competencies developed in educational robotics. This section has been reorganized to include intrapersonal skills such as creativity, autonomy, problem solving, autonomy and personal motivation. (Line 371-475) 

3.2.2. Interpersonal Skills and training competencies developed in educational robotics. This section encompasses collaborative work and skills to face the future with guarantees: to interact adequately with future professional and work contexts based on an increasingly technological society. (Line 476-547). 

For their part, in table 5 on line 311 (classification of variables) the variables are subdivided with respect to: Intrapersonal Skills and Interpersonal Skills. 

4) Lines 187-193 - here a point list was used, but there is only 1 research objective here. Is it justified in that case? 

We think so: There is a general objective: (now in line 206-212) 

- To investigate the perception of teachers of technology, robotics and/or programming and to study the social benefits of students' interpersonal and intrapersonal and/or academic skills for improving curriculum sustainability during the teaching-learning process, from the perspective of robotics and programming of students in Early Childhood Education, Primary Education, Compulsory Secondary Education and other educational levels in formal and non-formal education. 

And this is complemented with three specific objectives (line 214-225): 

  • To review and catalogue the academic teaching background, professional experience and evaluation experience of teachers of technology, programming and/or educational robotics.
  • To know the influence of educational robotics and programming in sustainable learning environments in school teaching and to analyze the material resources used by teachers of technology, programming and/or robotics during their teaching performance in the educational robotics classroom.  
  • To study the influence of robotics and/or educational programming on motivation in the teaching and learning process, collaborative work, the development of creativity and imagination, problem solving, personal autonomy and emotional management skills of students.  

5) In terms of research objective - is “the analysis” a research object in itself? The analysis should rather lead us to the goal, and not be an end in itself. 

It has been considered objective because of the importance of cataloging what type of professional (academic training, professional experience) makes up a teacher of educational robotics in the current educational context and evaluating in turn the experience in this subject and using the material resources presented below in this section of the results (3.1.2. Materials used by the teacher in the robotics classroom). 

The objective is poorly expressed, so its structure has been modified: 

- To review and catalogue the academic teaching background, professional experience and evaluation experience of teachers of technology, programming and/or educational robotics (line 215-217). 

6) After table titles (e.g. Table 5) there are 2 dots for some reason. 

Thank you for the observation: Corrected and eliminated 2 dotes in the statements of all tables. 

7) line 299 - after the number "59.1%" there are several unnecessary spaces. This situation also appears in other places (e.g. line 154) 

Thank you for the observation: Spaces have been corrected in all cases. 

Reviewer 2 Report

Comments and Suggestions for Authors

The article focused on using education robotics in the classroom to promote social sustainability. It is well-written and prepared. However, some points can be improved as follows:

1. The introduction is too long. This part should be shorter. It should just introduce the research work. All the contents regarding the explanation of the research background (the theoretical presentation of educational robotics and STEM ) should be written in one dedicated chapter, along with the description of the research aims and research questions.

2. Please check the size of the characters; some parts of the text don't have the same size (e.g., lines 199 to 203 and lines 361 to 366).

3. Lines 343 to 496 contain only a list of the teachers' opinions. Please categorize them based on the content analysis and summarize them. Therefore, there is no need to report all the teachers' sentences without marking the most relevant for the research objectives.

4. Please don't add another paragraph after the conclusion. Make sure that the description of your study's and research's limitations is included in the "Conclusions" paragraph. 

5. Please add more recent articles to the "references" to underline their relevance to modern teaching practices.

6. Please, check line 336; there is one space in between "of" and "creativity".

7. Please, check link 341, there is a typing mistake. 

Author Response

REVIEWER 2 

The changes introduced are indicated in grey color for better visualization: 

  1. The introduction is too long. This part should be shorter. It should just introduce the research work. All the contents regarding the explanation of the research background (the theoretical presentation of educational robotics and STEM ) should be written in one dedicated chapter, along with the description of the research aims and research questions.

The introduction has been reorganized and subdivided into four more specific and concise sections to make it easier to read and to avoid it being too long. More recent articles have also been added to the “references” to highlight their relevance to modern teaching practices as stated by another reviewer (highlighted in gray in the review document).    

1.1. Learning environments and Sustainable Development Goals 

1.2. Intrapersonal and interpersonal Skills in educational robotics classroom 

1.3. Students' motivation in educational robotics context 

1.4. Interdisciplinary curricular learning, innovation and material resources 

(Lines 34-193).

2. Please check the size of the characters; some parts of the text don't have the same size (e.g., lines 199 to 203 and lines 361 to 366).

Character size has been corrected in all cases.

3. Lines 343 to 496 contain only a list of the teachers' opinions. Please categorize them based on the content analysis and summarize them. Therefore, there is no need to report all the teachers' sentences without marking the most relevant for the research objectives.

The results of the research have been restructured, organizing the information in a more concrete way in each of the aspects dealt with, subdividing it into two more concise sections for better visualization (3.2.1. Intrapersonal Skills and training competencies developed in educational robotics and 3.2.2.) Each of the items in these sections (creativity, personal autonomy, collaborative work...) are structured in an individualized way, showing detailed results and then the teachers' answers in each of them. Complementing this, two tables summarizing and highlighting the key contributions (most relevant teacher responses) of each of these sub-sections are presented in tables 6 and 7.) (Lines 314-547).

Added two tables to synthesize the most important responses of the teachers in the study in each of the intrapersonal and interpersonal skills:   

  • Table 6 (line 473). Most relevant information collected from the sample of technology, programming and/or robotics teachers on the intrapersonal skills developed in the educational robotics classroom  
  • Table 7 (line 544). Most relevant information collected from the sample of technology, programming and/or robotics teachers on the interpersonal skills developed in the educational robotics classroom. 
  1. Please don't add another paragraph after the conclusion. Make sure that the description of your study's and research's limitations is included in the "Conclusions" paragraph. 

The description of the study's and research's limitations is included in the "Conclusions" paragraph (lines 747-762).    

  1. Please add more recent articles to the "references" to underline their relevance to modern teaching practices.

More recent articles have also been added to the references to highlight their relevance to modern teaching practices as stated by another reviewer (highlighted in gray in the review document):  

- Reference 6: lines 63-66. 

- Reference 29 and reference 30: lines 172-178. 

  1. Please, check line 336; there is one space in between "of" and "creativity".

Thank you for the observation: Corrected the error. 

  1. Please, check link 341, there is a typing mistake.

Thank you for the observation: The typing mistake has been corrected. 

Reviewer 3 Report

Comments and Suggestions for Authors

The entire Introduction reads as a series of disjointed, short paragraphs about individual studies/papers. This section would be improved by synthesizing the background literature into a coherent argument that supports the study. Right now, it is more of an annotated bibliography, or essentially a list of studies that may be relevant.

Line 182: the authors say “the proposal of the study.” If it is the research question for the current study, state it that way. By phrasing as you have, it is unclear whether this is research that has already been conducted or if it is a proposal for research to be conducted in the future.

The results are a bit hard to read the way they’re arranged. They read a bit like the introduction, where it seems like a list of ideas rather than a coherent section of text. It would be nice to better incorporate the quotes with the other text, rather than listing related quotes after each section of text.

When discussing the teachers’ degrees, do the people with education degrees only have education degrees? Or do some also have other (which could be relevant to robotics or not) degrees as well? I would expect teachers of younger grades to perhaps only have education degrees, though teachers of older grades may also have other degrees.

I’m not clear on where the conclusions about interpersonal and intrapersonal skills comes from.

There is a lot of interesting data here, but as written, it was hard to follow in a way that led me to understand the big impacts of this study. Focus revisions on telling that story coherently. Synthesize the background literature to illustrate the gap in the literature that points to the need for the study, present your data, and connect it back to the literature to show how your study is adding to the body of knowledge. Focusing on that as a coherent story will make this easier to read and digest.

Comments on the Quality of English Language

There are typos and some odd word choices, but it is mostly good.

Author Response

REVIEWER 3 

The changes introduced are indicated in grey color for better visualization: 

1. The entire Introduction reads as a series of disjointed, short paragraphs about individual studies/papers. This section would be improved by synthesizing the background literature into a coherent argument that supports the study. Right now, it is more of an annotated bibliography, or essentially a list of studies that may be relevant. 

The introduction has been reorganized and subdivided into four more specific and concise sections to make it easier to read and to avoid it being too long. More recent articles have also been added to the “references” to highlight their relevance to modern teaching practices as stated by another reviewer (highlighted in gray in the review document).   

1.1. Learning environments and Sustainable Development Goals 

1.2. Intrapersonal and interpersonal Skills in educational robotics classroom 

1.3. Students' motivation in educational robotics context 

1.4. Interdisciplinary curricular learning, innovation and material resources 

(Lines 34-193). 

2. Line 182: the authors say “the proposal of the study.” If it is the research question for the current study, state it that way. By phrasing as you have, it is unclear whether this is research that has already been conducted or if it is a proposal for research to be conducted in the future. 

The way of expressing the research question has been modified and redefined (line 200): The research question is: 

3. The results are a bit hard to read the way they’re arranged. They read a bit like the introduction, where it seems like a list of ideas rather than a coherent section of text. It would be nice to better incorporate the quotes with the other text, rather than listing related quotes after each section of text. I’m not clear on where the conclusions about interpersonal and intrapersonal skills comes from. 

The results of the research have been restructured, organizing the information in a more concrete way in each of the aspects dealt with. Each of the items in these sections (creativity, personal autonomy, collaborative work...) are structured in an individualized way, showing detailed results and then the teachers' answers in each of them. Complementing this, two tables summarizing and highlighting the key contributions (most relevant teacher responses) of each of these sub-sections are presented in tables 6 and 7.)  (Lines 314-547).

It is advisable to differentiate part of the results provided in point 3.2. Knowledge results and teaching involvement, so this section has been subdivided into two separate sections:  

- 3.2.1. Intrapersonal Skills and training competencies developed in educational robotics. This section has been reorganized to include intrapersonal skills such as creativity, autonomy, problem solving, autonomy and personal motivation. (Line 371-475). 

- 3.2.2. Interpersonal Skills and training competencies developed in educational robotics. This section encompasses collaborative work and skills to face the future with guarantees: to interact adequately with future professional and work contexts based on an increasingly technological society. (Line 476-547). 

For their part, in table 5 on line 311 (classification of variables) the variables are subdivided with respect to: Intrapersonal Skills and Interpersonal Skills. 

Added two tables to synthesize the most important responses of the teachers in the study in each of the intrapersonal and interpersonal skills:  

  • Table 6 (line 473). Most relevant information collected from the sample of technology, programming and/or robotics teachers on the intrapersonal skills developed in the educational robotics classroom 
  • Table 7 (line 544). Most relevant information collected from the sample of technology, programming and/or robotics teachers on the interpersonal skills developed in the educational robotics classroom. 

4. When discussing the teachers’ degrees, do the people with education degrees only have education degrees? Or do some also have other (which could be relevant to robotics or not) degrees as well? I would expect teachers of younger grades to perhaps only have education degrees, though teachers of older grades may also have other degrees. 

It is an interesting aspect to deal with as you state, although in the Carrera and Ferrán (2017) questionnaire that was used to collect the information the possibility of selecting several degrees or “other” studies is given, the teachers in most cases have responded concisely in a single type of academic study. In future studies, the interest in collecting more information regarding master's and doctoral studies, university courses (own degrees) or doctorates can be accentuated.... Thank you very much for the recommendation but it is a pity that on this occasion I cannot draw conclusions in that sense. 

5. There is a lot of interesting data here, but as written, it was hard to follow in a way that led me to understand the big impacts of this study. Focus revisions on telling that story coherently. Synthesize the background literature to illustrate the gap in the literature that points to the need for the study, present your data, and connect it back to the literature to show how your study is adding to the body of knowledge. Focusing on that as a coherent story will make this easier to read and digest. 

Thank you for your recommendation. We have tried to reorginized an link the introduction to the results, discussion and conclusions in a more coherent way. 

First, the introduction and the theoretical framework have been reorganized and subdivided into the four sub-sections described above, joining the information in a more coherent way, modifying the wording to make it easier to read.  

Secondly, the results of the research have been restructured, organizing the information in a more concrete way in each of the aspects dealt with, subdividing it into two more concise sections for better visualization (3.2.1. Intrapersonal Skills and training competencies developed in educational robotics and 3.2.2.) Each of the items in these sections (creativity, personal autonomy, collaborative work...) are structured in an individualized way, showing detailed results and then the teachers' answers in each of them. Complementing this, two tables summarizing and highlighting the key contributions (most relevant teacher responses) of each of these sub-sections are presented in tables 6 and 7.) 

Finally, the discussion and conclusions have been reorganized and better implemented for a better linkage with the theoretical framework and the results presented above. 

Round 2

Reviewer 3 Report

Comments and Suggestions for Authors

Thank you for addressing the concerns. 

Comments on the Quality of English Language

There are still a couple of places with some phrasing and/or grammar issues, but the paper is readable as far as the English language is concerned.